# Using Multi-Temporal MODIS NDVI Data to Monitor Tea Status and Forecast Yield: A Case Study at Tanuyen, Laichau, Vietnam

**Phamchimai Phan** [1,2]**, Nengcheng Chen** [1,3,]*****, Lei Xu** [1] **and Zeqiang Chen** [1]

1   State Key Laboratory of Information Engineering in Surveying, Mapping, and Remote Sensing, Wuhan University, Wuhan 430079, China; phanphamchimai87@gmail.com (P.P.); xulei123@whu.edu.cn (L.X.); ZeqiangChen@whu.edu.cn (Z.C.)
2   The Faculty of Tourism, Thai Nguyen University of Science, Thai Nguyen 250000, Vietnam
3   Collaborative Innovation Center of Geospatial Technology, Wuhan 430079, China
*   Correspondence: cnc@whu.edu.cn; Tel.: +86-027-68779996

**Abstract:** Tea is a cash crop that improves the quality of life for people in the Tanuyen District of Laichau Province, Vietnam. Tea yield, however, has stagnated in recent years, due to changes in temperature, precipitation, the age of the tea bushes, and diseases. Developing an approach for monitoring tea bushes by remote sensing and Geographic Information Systems (GIS) might be a way to alleviate this problem. Using multi-temporal remote sensing data, the paper details an investigation of the changes in tea health and yield forecasting through the normalized difference vegetation index (NDVI). In this study, we used NDVI as a support tool to demonstrate the temporal and spatial changes in NDVI through the extract tea NDVI value and calculate the mean NDVI value. The results of the study showed that the minimum NDVI value was 0.42 during January 2013 and February 2015 and 2016. The maximum NDVI value was in August 2015 and June 2017. We indicate that the linear relationship between NDVI value and mean temperature was strong with $R^2 = 0.79$ Our results confirm that the combination of meteorological data and NDVI data can achieve a high performance of yield prediction. Three models to predict tea yield were conducted: support vector machine (SVM), random forest (RF), and the traditional linear regression model (TLRM). For period 2009 to 2018, the prediction tea yield by the RF model was the best with a $R^2 = 0.73$, by SVM it was 0.66, and 0.57 with the TLRM. Three evaluation indicators were used to consider accuracy: the coefficient of determination ($R^2$), root-mean-square error (RMSE), and percentage error of tea yield (PETY). The highest accuracy for the three models was in 2015 with a $R^2 \geq 0.87$, RMSE < 50 kg/ha, and PETY less 3% error. In the other years, the prediction accuracy was higher in the SVM and RF models. Meanwhile, the RF algorithm was better than PETY ($\leq 10\%$) and the root mean square error for this algorithm was significantly less ($\leq 80$ kg/ha). RMSE and PETY showed relatively good values in the TLRM model with a RMSE from 80 to 100 kg/ha and a PETY from 8 to 15%.

**Keywords:** NDVI; tea monitoring; yield forecasting; remote sensing; support vector machine (SVM); random forest (RF)

## 1. Introduction

The northwest Vietnam region featured mountainous terrain categories with the temperature, water, and light being suitable conditions for tea production. Laichau is one of the main tea growing areas in the northwest region of Vietnam where Tanuyen District is the key tea producing area. It has 2854 ha of tea plants, accounting for two-thirds of the tea-producing area of the whole province. Tea is a crop that does not compete with food crops; tea plantations cover bare land, bare hills, and prevent

erosion. At the same time, tea is a cash crop that improves the quality of life for people in the Tanuyen District of Laichau Province. Each family earns an average of 100–120 million Viet Nam Dong per year from tea trees. Tea products are mainly exported to countries and regions like Pakistan, India, China, and Taiwan. It can be said that tea is a crop that brings economic benefits and creates jobs for people in Vietnam, and accurate estimations of tea yield are becoming increasingly important. In particular, the yield estimation of tea may play an important role in supporting policy building and decision-making in agriculture, even affecting the agricultural economic development of the region. Tea yield, however, has stagnated in recent years, due to changes in temperature, precipitation, the age of the tea bushes, and diseases [1]. In this study, we will mainly focus on the meteorological factors and Normalized difference vegetation index (NDVI), which are the most important factors affecting tea yield. Remote sensing and GIS were combined to monitor tea bush status and forecast tea yield based these factors.

The application of remote sensing technology for vegetation condition monitoring has been used extensively during the past several decades, providing a timely estimation in the growth and development changes of crops [2,3]. The status and yield crop can be used as early forecasting to provide up-to-date information before the crop harvest period by the crop monitoring system. Numerous previous research has shown that NDVI is a widely used spectral transformation method and helpful crop monitoring tool [3–8]. A higher NDVI value appears in healthy and dense plants, and a lower NDVI appears in sparse plants, which means that stressed plants are displayed as dark areas amongst the brighter, healthier crop areas [9,10]. R.M.S.S. Rajapakse and Dutta [9,10] demonstrated that areas of consistently healthy and vigorous crops appeared uniformly bright, and that stressed vegetation appears dark amongst the brighter, healthier crop area. They assessed the tea bush through texture, tonal variation, and NDVI values, respectively; however, the temporal analysis was limited. To better understand the health of tea over a long time and the difference from previous research, monitoring the health of tea in Tanuyen should be undertaken.

Remote sensing and GIS are considered as tools for the detection and identification of tea classification [11–13] and the monitoring of tea plantations through the spectral characteristics of tea plants [9] to map and monitor the tea plantation impact on land-use/land-cover [14]. Tea bush health has been assessed using both texture and tonal variations [10]. The diseased patches were delineated using both texture analysis and the classified based images. The percentage of healthy tea was showed with three thresholds: under healthy tea, moderately affected tea, and diseased tea by LANDSAT images from December 2001, an ASTER image from June 2004, and LISS III image from February 2004. Barman [6] detected and showed the tea health status using three thresholds: "healthy", "moderately healthy", and "unhealthy" by investigating and analyzing the spectral responses in remote sensing images.

In order to estimate crop yield, some yield forecasting models have been carried out for some crops such as rice, sugarcane, wheat, grain, and potato. All of these studies used NDVI as a support tool for the accurate and efficient estimation of crop yield [15–17], which is inspected as a productivity measure and is sensitive to parameters of vegetation such as the green leaf area index, the fraction of absorbed photo synthetically active radiation, and the ground surface percentage covered by vegetation [18].

These studies have mainly focused on NDVI imagery from the Landsat satellite to monitor the development of the biomass of rice crop such as that by Pandey [19]. Dong [20] demonstrated that integral NDVI had a close relationship with winter wheat yield. Paul c. Doraiswamy et al. [2] used a Landsat classification of spring wheat for North Dakota, and a crop mask was developed to help retrieve the NDVI values primarily for spring wheat crop. The relationship between NDVI and above ground biomass (AGB) was established and then the harvest index (HI) was calculated based on the change of NDVI from the period of re-greening to maturity [15]. Foster et al. [21] determined the relationships between biomass yield and canopy height and canopy NDVI under different cropping systems. S. K. Bala [16] used eight days of TERRA MODIS reflectance data to estimate the yield of potato in the Munshiganj area of Bangladesh. In other research, an empirical ordinary least squares regression model was proposed for yield estimation at provincial and national levels by RiadBalaghi [17].

NDVI has been used for tea yield prediction in the last decade. Dutta [22] mentioned the prediction of green leaf tea yield and carried out tea yields based on the image-derived leaf area index (LAI). To further check the relationship between MODIS NDVI and LAI, an empirical equation was established. However, it was found that the NDVI observations at different time periods could not interpret much contradiction in tea leaf yield. Dutta [10] tested whether MODIS derived NDVI was related to LAI and an empirical equation was established that showed that LAI in tea had a significant and linear relationship with NDVI. Empirical relations between the NDVI and crown density were investigated, whereas the relation between the crown density and tea yield remains to be investigated by F Fauziana [23]. In order to predict tea yield, the statistical relationship was established between climate variables, which were regarded as predictor parameters and tea yield (predictand), by [24–28]. All studies used meteorological data, or satellite image data, or the incorporation of these two to establish empirical models for tea yield prediction. However, performances of yield prediction using empirical methods mostly use the traditional linear regression model (TLRM). Machine learning is also an efficient empirical method for classification and prediction. Jaikla et al. [29] estimated rice yields using SVM (support vector machine). Kim and Nari [30] described the corn yield estimation in Iowa State using four machine learning approaches such as SVM (support vector machine), RF (random forest), ERT (extremely randomized trees) and DL (deep learning). Narayanan Balakrishnan [31], Priya [32], and Arun Kumar [33] proposed a crop yield prediction model such as the SVM classification technique. Yaping Cai et al. [34] used three mainstream machine learning methods (SVM, RF, and neural network) to build various empirical models for yield prediction. The random forest algorithm was also considered as a strong tool for predicting the yield of the crop. To improve the prediction accuracy of tea yield, our work proposes a prediction model for yield forecast based on support vector machine (SVM) and random forest (RF) to replace the traditional linear regression model (TLRM). Furthermore, comparisons of the validation statistics among them are presented.

## 2. Materials and Methods

### 2.1. Materials

This study data can be categorized into the following three types: earth observation data, field survey data, and data collected at Tanuyen District (Table 1). Earth observation data were collected including Sentinel-2 images and Moderate-Resolution Imaging Spectroradiometer (MODIS) NDVI images. Two Sentinel-2 images were taken on 3 November 2018 at a 10-m resolution to cover the study area. MODIS NDVI products were collected from 1 January 2009 to 31 December 2018, for a total of 120 images. The ground truth data collection was used for image classification and validation. The GPS was used for taking the coordinates of the area from where the reading was collected. Tea yield data and base map data were collected with high reliability from the Department of Natural Resources and Environment, Tanuyen District. In addition, meteorological data were collected from the hydro-meteorological station of Laichau Province, which supplies the aggregated data of all the stations in the province such as Lai chau town Station, Tam duong Station, Muong te Station, Sin ho Station, and Thanuyen Station. Meanwhile, meteorological data from Thanuyen Station were used for monitoring health and forecasting tea yield as its position is nearest to Tanuyen's tea gardens and Tanuyen's area comes from southern half of Thanuyen District.

**Table 1.** Data sets for land-cover map and tea yield forecasting.

| Data Set | Source | Characteristics/Features | Date |
|---|---|---|---|
| Sentinel-2 | https://earthexplorer.usgs.gov | 13 spectral bands: four bands at 10 m, six bands at 20 m and three bands at 60 m spatial resolution (images) | 3 November 2018 |

**Table 1.** *Cont.*

| Data Set | Source | Characteristics/Features | Date |
|---|---|---|---|
| MODIS NDVI | MODIS13A3H27V06 (Laichau) | MODIS/Terra vegetation Indices Monthly L3 Global 1 km (images) | January 2005 to December 2018 |
| Climate data | Hydro-meteorological station of Laichau Province [35] (Thanuyen station: 21°57′N and 103°53′E) | Mean temperature, minimum temperature (Tmin), maximum temperature (Tmax), precipitation, solar radiation | January 2005 to December 2018 |
| Tea yield data | Department of Natural Resources and Environment, Tanuyen District, Laichau Province, Vietnam [1] | Yield monitor data (productivity/area), unit (ton/ha) | January 2009 to December 2018 |
| Base map data | Department of Natural Resources and Environment, Tanuyen District, Laichau Province, Vietnam | Vector | 2009 |
| Field survey data | GPS (GTField) | Longitude, latitude | November 2018 |

### 2.1.1. Sentinel-2

Sentinel-2 is the second generation earth observation (EO) satellite operated by the European Space Agency (ESA) [36]. Sentinel-2A and Sentinel-2B were launched in June 2015 and March 2017, respectively, as an integral part of Europe's Copernicus program aiming at independent and continued global observation capacities [37]. The quality report of Sentinel-2 data in 2018 [38] reported that Sentinel-2A images before 15 June 2016 had registration errors due to three main contributors: (a) dynamic vibration residuals mainly related to on-board oscillations; (b) static Light Of Sight calibration residuals; and (c) correlation noise and outliers. In 2017, Sentinel-2 images had to be discarded due to cloud coverage in this study area. Therefore, Sentinel-2 images in 2018 were selected. Sentinel-2 images have a high spatial resolution, so Jun Zhu et al. [13] and Vanessa Paredes Gómez [39] used them to test the classification ability. Meanwhile, Jun Zhu [13] identified tea plantations based on multi-temporal Sentinel-2 images and a multi-feature random forest (RF) algorithm. In this research, Sentinel-2 images were used on 3rd November 2018 with title numbers T48 QTK and T48 QUK to cover the study area. Two images had a small fraction of cloud (<5%) cover over this area, thus helping in the easy interpretation of the objects. These images were used for land-use/land-cover (LULC) classification.

### 2.1.2. Moderate Resolution Imaging Spectroradiometer Vegetation Index Data

The MODIS NDVI complements National Oceanic and Atmospheric Administration's Advanced Very High-Resolution Radiometer (AVHRR) NDVI produces and provides continuity for historical applications. The Terra MODIS Vegetation Indices (MOD13A3) Version 6 data are provided monthly at 1-km spatial resolution as a gridded Level 3 product in the sinusoidal projection. In generating this monthly product, the algorithm ingests all the MOD13A2 products that overlap the month and employs a weighted temporal average. Based on the application of remote sensing techniques, the vegetation indices obtained from canopies were developed by scientists to estimate and assess the vegetation cover and growth dynamics [40]. The vegetation index plays a supporting role in determining the distribution of vegetation. The relative density and vegetation health are displayed on the basis of each pixel or in the satellite image through vegetation indices. NDVI is one of the most widely used vegetation indices for monitoring vegetation stress [8].

### 2.1.3. The Other Data

The other data that were used for this study include the meteorological data (mean temperature, Tmin, Tmax, solar radiation, precipitation), base map data, the tea yield data, and the field survey

data. The meteorological data were collected from the hydro-meteorological station office of Laichau Province [35]. The base map data provide the information on the location such as the administrative map, river map, and residential map. Tea yield was supplied by monitoring total productivity of tea (ton) after harvest time from the tea gardens following the unit area. The field data were collected through survey and measurements at the fieldwork such as the longitude and latitude of some points of the tea and cropland area. The survey data were collected by using a GPS (GTField) instrument to take coordinates of the tea and cropland. Before 2009, Tanuyen District with Thanuyen District formed one district that were separated to become two administrative units in 2009. Thus, in this research, we only collected the tea yield data from 2009 to the present.

### 2.1.4. Study Area

Tanuyen is a rural district of Laichau Province in the northwest region of Vietnam (Figure 1). It is a new district and was established in 2008 and its area encompasses the southern half Thanuyen District. Tanuyen has nine communes, with an area of 903.27 km². Located on the western side of the Hoang Lien Son range, Tanuyen District in Laichau Province has quite ideal conditions to develop tea plants. Being a plateau, the altitude is 600–700 m above sea level. Tanuyen has many long mountain ranges interspersed with narrow rivers. The climate has clear rainy and dry seasons. The annual average temperature is about 22 °C and the rainfall is 1800–2300 mm/year. The soil composition is mainly red and yellow, suitable for the development of perennial crops, especially tea trees. The main tea varieties here are Shan Tuyet tea, Kim Tuyen team and Thanh Tam tea. Tea is grown in many communes including Phuc Khoa, Muong Khoa, Than Thuoc, Trung Dong, Pac Ta, and Tanuyen town.

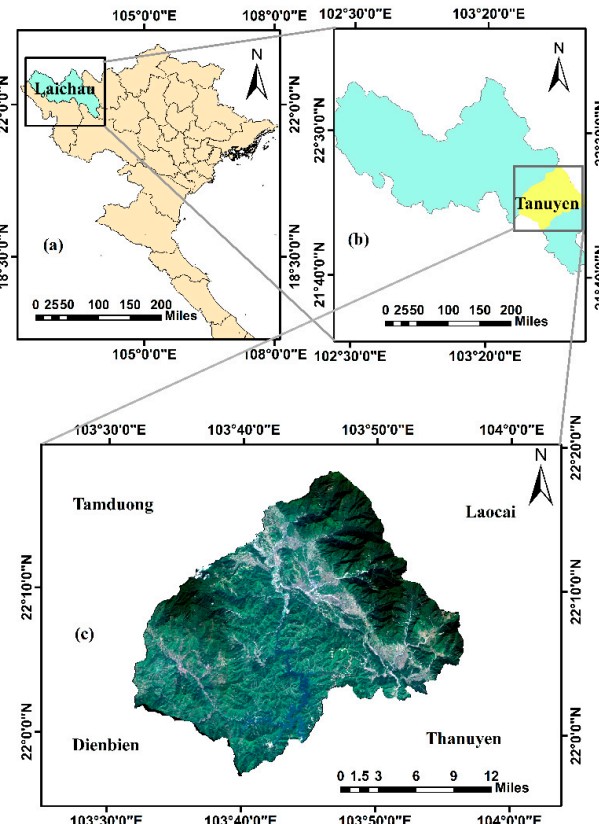

**Figure 1.** The location of Tanuyen District in Laichau, Vietnam. (**a**) The location of Laichau Province in Vietnam; (**b**) the location of Tanuyen District in Laichau Province; and (**c**) the Tanuyen District observed from Sentinel-2 image on 3 November 2018.

## 2.2. Methods

In this study, before classifying the Sentinel-2 images, the layers must be stacked, broken into subsets, and geometrically corrected. The MODIS NDVI products were geometrically corrected and then combined with tea boundaries to obtain the tea NDVI data. To forecast tea yield, we approached two mainstream machine learning methods, support vector machine (SVM) and random forest (RF), and then compared their results with the traditional linear regression model (Figure 2).

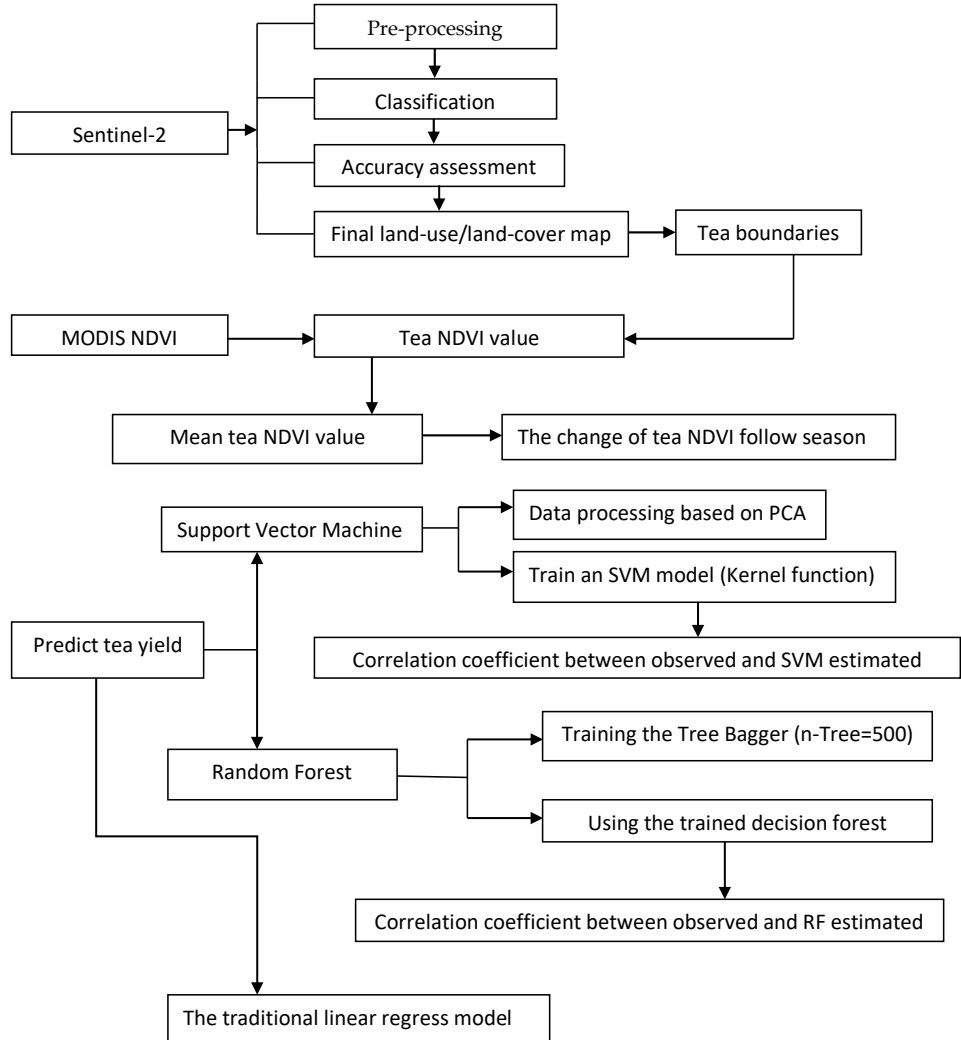

**Figure 2.** The proposed methods to monitor tea health and forecast tea yield.

### 2.2.1. Land-Use/Land-Cover Map

All steps for image pre-processing were carried out by ERDAS IMAGE 2015 software. The next step was to classify the image into six classes: tea, forest, cropland, water-body, settlement, and barren land to separate the tea boundaries. In order to detect tea patches, the temporal images were visually interpreted. The false-color composite (FCC) for Sentinel-2 was generated at 10:8:2 respectively. As shown from the FCC and texture of each object, tea appears as a green and yellow-green colors; cropland has magenta and pink colors; water is black; forest is yellow, red, and black-green colors; settlement has slate-blue and maroon colors; barren land is light-green and whitish-green in color. Unsupervised and supervised classification are two major categories that are widely used in image classification techniques. In this study, supervised image classification was applied using maximum likelihood to classify the various pixel values or spectral signatures that should be associated with

each class. A total of 620 sample points were used for the training dataset and each class contained 100 sample points, except for barren land and crop land. The algorithm used the spectral signatures from these training areas to classify the whole image. The classified images were validated with the ground truth data and Google Earth. In Google Earth, the tea bushes appear with a green color and raw texture and cropland appears with a grey green, smooth texture. Maximum likelihood classification (MLC) gave the best results compared to the minimum distance to mean (MDM) and parallelepiped classification. Post classification was carried out with clump and sieve to smooth and remove small groups of pixels from the classification.

### 2.2.2. Calculating the Mean Normalized Difference Vegetation Index (NDVI) Value of Tea

The health state of the vegetation was estimated through NDVI, which is considered as a qualitative and quantitative measure of vegetation cover by the reflected light of the vegetation at certain frequencies. Chlorophyll (a health indicator) strongly absorbs visible light, and the cellular structure of the leaves strongly reflect near-infrared light. The healthy vegetation will reflect more near-infrared energy than stress vegetation.

$$NDVI = \frac{NIR - R}{NIR + R} \tag{1}$$

NDVI observations at different time periods can show healthy vegetation conditions. The NDVI of tea was found from the extracted NDVI of Laichau Province and the tea boundaries.

The aggregation process of NDVI is presented as follows. First, the data (MODIS13A3H27V06) were downloaded, image processing was carried out, and then resampled to a $10 \times 10$ m spatial resolution. Then, all NDVI values were further masked based on the tea planting areas. Finally, we calculated the averages of the NDVI from the pixels within each point representing a tea plantation in each month. Data processing was mainly carried out on ArcGIS and MATLAB software

Based on the United States Geological Survey (USGS) [41], we proposed the class names of "moderate" and "healthy" for monitoring the tea NDVI status, where a value between 0.4 and 0.59 is moderate and ≥0.6 is healthy. The linear regression analysis was carried out using the NDVI and mean temperature. NDVI was considered as a dependent factor and temperature was considered as the independent factors.

### 2.2.3. Predicting Tea Yield Methods

To forecast tea yield, two mainstream machine learning methods were used: support vector machine (SVM) and random forest (RF). The traditional linear regression model (TLRM) was also established by the same data to compare the accuracy of TLRM, SVM, and RF and a scatter plot of the monthly observed and predicted tea yield at Tanuyen was derived using SVM, RF, and TLRM. The performances of SVM, RF, and TLRM were compared in the study to select the best method to predict tea yield in Tanuyen.

Support vector machine (SVM) is a machine learning method used for classification purposes, regression analysis, and time series prediction. SVM analysis is a popular machine learning tool for classification and regression, first identified by Vladimir Vapnik [42]. SVM is a machine learning technique that is capable of capturing the highly nonlinear relationship between the predictor and predict and thus performs better than conventional linear regression models [43]. SVM can also solve nonlinear regression problem based on kernel functions. In SVM regression, the set of training data includes predictor variables and observed response values. The goal is to find a function $f(x)$ that deviates from $y_n$ by a value no greater than $\varepsilon$ for each training point $x$, and at the same time, is as flat as possible. In this situation, the predictor variables include mean temperature, Tmin, Tmax, precipitation, solar radiation, and NDVI, which are considered as the input parameters. SVM is a kernel based learning algorithm where the key is kernel function, support solver SMO, and the number

observations of this study is 100. The Gaussian radial basis function (RBF) is one of the most popular kernel functions.

$$y = f(x) = \{\sum_{i=1}^{N} (\alpha_i K(x_i, x)\} - b)\qquad(2)$$

where K denotes the kernel function; $\alpha_i$ and b are the parameters; N is the number of training data; $x_i$ is the vector used in the training process; and $x$ is the predictor vector. The parameters $\alpha_i$ and b are derived by maximizing their objective function.

$$K(x_i, x) = \exp\{- g\|x_i, x\|\}2\qquad(3)$$

where g denotes the kernel parameter to measure the width of the kernel function in RBF.

Random forest (RF) is an ensemble algorithm introduced by Breiman [44], which relies on the number of tree decisions for classification and regression analysis [45]. It is a non-parametric statistical regression algorithm that produces numerous independent trees to find a final decision through two randomization approaches, one in the selection of the training samples and the other in the selection of variables at each node of a tree. The number of tree (ntree) samples are bootstrapped from the training data, then a tree is grown and split according to the predictors. The number of trees is trained, then used in classification or regression. The ensemble process in deciding the final class of ntree is obtained by a vote method. The advantage of the RF model is capable of capturing non-linear interaction between the features and the target.

This paper focused on the yield prediction of tea using the existing data and relying on the use of RF and SVM algorithms. The SVM and RF models were calibrated and validated during the period 2009 to 2018. The accuracy of the SVM and RF models are dependent on the selected model parameters. The tea yield was predicted from the available data as weather factor monthly (mean temperature, Tmin, Tmax, precipitation, solar radiation), NDVI, and historic tea yield.

To assess the results of the SVM algorithms, calibration and cross-validation were carried out. In the calibration assessment, the training set and validation were used to develop the same model. In this paper, the size of the training dataset was 100 values, occupying 83%, and 20 values were testing data, occupying 17%. This 17% of data was used to check the validation of the model accuracy. The prediction results were obtained via an external validation by training and testing the models with the training and test datasets, respectively.

SVM and RF have been used successfully in many studies to predict crop yield, but have not been used for the development of a tea yield estimation model. In this work, we built a prediction model based on the SVM and RF algorithm.

We adopted the root-mean-square error (RMSE), the coefficient of determination (R2), and percentage errors of tea yield in different year (PETY) to evaluate the performance of the models, which can be calculated as follows:

$$RMSE = \sqrt{\frac{\sum_{i=1}^{n}(x_i - y_i)^2}{n}}\qquad(4)$$

$$PETY = \frac{|x_i - y_i|}{|y_i|} * 100\qquad(5)$$

where $n$ (i = 1, 2, ... ,n) is the number of samples used for the SVM, RF, and TLRM model; $x_i$ is the observed tea yield; and $y_i$ is the predicted tea yield.

RMSE is an important statistical calculation that is used to measure the error between two datasets. A smaller RMSE value indicates a small error, while a large RMSE shows a large error between the predicted values and observed value. The percent error of tea yield in different year is useful tool for determining the precision of prediction result. Error bar represents one standard error.

## 3. Results

### 3.1. Landuse–Landcover Classification and Normalized Difference Vegetation Index (NDVI) Analysis

#### 3.1.1. Land-Use/Land-Cover Map and the Change of Tea NDVI

The Sentinel-2 image collected on 3 November 2018 was used for land-use/land-cover (LULC) classification. The image was classified into six classes: tea, forest, cropland, settlement, water-body and barren-land. Supervised classification was performed using the maximum likelihood classifier to classify the image. In this research, LULC was described as follows: the cropland class included rice, corn, and vegetables. The forest class included primary forest, regenerated forest, and plantation forest. The barren land class included grass, shallow river, shrubs, etc. The classified map is shown in Figure 3. The classified images were validated with the ground truth data and Google Earth.

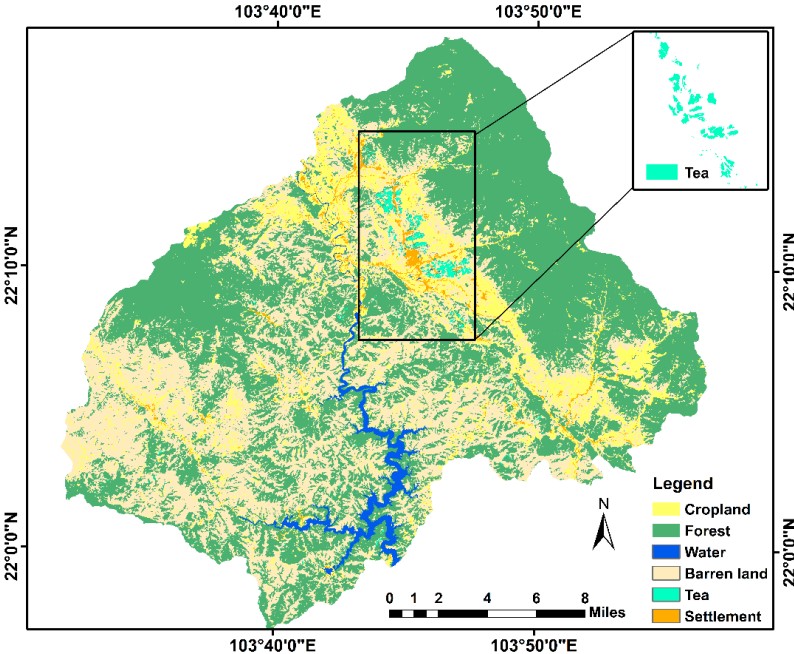

**Figure 3.** Land-use/land-cover map and tea boundaries of Tanuyen, Laichau.

The assessment of accuracy is a method to quantify the preciseness of the classifieds image by constructing an "error matrix", as shown in Table 2. The matrix is filled with counts taken from a sample. The utility added random points will generate random points throughout the classified image. The reference dataset created was also taken with the help of high-resolution Google Earth images. All of the reference data were collected and the images can generate an accuracy assessment report. After the points are generated, these reference values are compared to the class value of the classified image. The overall classification accuracy was found to be 95% and the Kappa coefficient was 94%.

**Table 2.** Error matrix for the supervised classification of the intermediate study area.

| Class Name | Sentinel-2 | |
| --- | --- | --- |
| | Producer Accuracy | User Accuracy |
| Tea | 99% | 100% |
| Cropland | 100% | 98% |
| Forest | 97.03% | 97.03% |
| Settlement | 85% | 100% |
| Water | 100% | 98% |
| Barren land | 100% | 81% |

For the class-specific accuracy, tea, cropland, water-body, and forest achieved the highest accuracy of 97%. Settlement class had a producer accuracy of 85%, likely because the settlements in this area have living habitats scattered in the mountain and valley in a very small area. All of the settlement pixels had a lot of color like red, blue, and light ocean color, so some pixels were mixed with barren land. The results of this study show that Sentinel-2 has great potential for crop classification with accurate results.

The spatial coverage of the area is small, but brings great economic profit for the citizen. Based on the classification result, it is clear that forest, shrub, and barren land made up the largest area, which can be explained by the fact that this is a mountainous district in the province. The fraction of tea plants is the second largest in the district's crop structure, estimating for 25% of the agricultural land area, but as tea is a crop with high economic efficiency, we focused on its development.

### 3.1.2. The Change of Tea NDVI on Temporal and Spatial

The NDVI has a strong relationship with the biophysical parameters of plants. Previous study showed that the NDVI value can be successfully used in the dynamic and trend assessment of long-term vegetative states [46] to detect vegetation and climate interactions [47]. The NDVI observations at different time periods can also show crop health condition. Tea boundaries were identified through classifying the Sentinel-2 images. The NDVI values of all tea gardens were extracted from the tea boundaries and MODIS NDVI each month. Then, the mean NDVI values of tea (Table 3) were obtained through aggregating the pixels within each polygon representing tea plantations.

**Table 3.** The monthly mean NDVI value of tea in Tanuyen District.

| Year | 1 | 2 | 3 | 4 | 5 | 6 | 7 | 8 | 9 | 10 | 11 | 12 |
|------|------|------|------|------|------|------|------|------|------|------|------|------|
| 2009 | 0.50 | 0.53 | 0.43 | 0.68 | 0.69 | 0.72 | 0.78 | 0.77 | 0.74 | 0.65 | 0.60 | 0.54 |
| 2010 | 0.51 | 0.48 | 0.47 | 0.68 | 0.64 | 0.72 | 0.77 | 0.76 | 0.73 | 0.66 | 0.65 | 0.57 |
| 2011 | 0.46 | 0.48 | 0.45 | 0.68 | 0.71 | 0.74 | 0.76 | 0.77 | 0.74 | 0.70 | 0.62 | 0.56 |
| 2012 | 0.45 | 0.53 | 0.48 | 0.65 | 0.71 | 0.74 | 0.75 | 0.78 | 0.75 | 0.73 | 0.67 | 0.59 |
| 2013 | 0.42 | 0.45 | 0.66 | 0.69 | 0.70 | 0.77 | 0.75 | 0.78 | 0.69 | 0.65 | 0.61 | 0.45 |
| 2014 | 0.44 | 0.46 | 0.66 | 0.69 | 0.73 | 0.75 | 0.78 | 0.78 | 0.74 | 0.70 | 0.65 | 0.57 |
| 2015 | 0.55 | 0.42 | 0.66 | 0.68 | 0.71 | 0.75 | 0.78 | 0.80 | 0.75 | 0.72 | 0.69 | 0.62 |
| 2016 | 0.58 | 0.42 | 0.46 | 0.71 | 0.68 | 0.73 | 0.79 | 0.79 | 0.73 | 0.68 | 0.62 | 0.49 |
| 2017 | 0.43 | 0.46 | 0.67 | 0.68 | 0.68 | 0.80 | 0.78 | 0.77 | 0.74 | 0.71 | 0.66 | 0.59 |
| 2018 | 0.45 | 0.50 | 0.47 | 0.70 | 0.70 | 0.78 | 0.78 | 0.78 | 0.74 | 0.71 | 0.66 | 0.65 |

In this study, we used 120 images of MODIS NDVI products and demonstrated the change of NDVI in the temporal and spatial analysis from 2009 to 2018. Some tea NDVI maps were established in January and June to show the change in tea NDVI for the season. From the map below, it is easy to observe the difference in tea NDVI value between each season by establishing a tea NDVI map using January and June, as shown in Figure 4a,c. In January, the NDVI value was approximately 0.3 to 0.5, when it is winter in Vietnam. At this time, the tea NDVI obtained the minimum value, even though it only obtained a 0.2 value in some gardens. The June NDVI value was observed to be incremented, and had values of around 0.6 to 0.8, which coincided with the summer in Tanuyen. The mean NDVI value of tea was calculated for January and June from 2009 to 2018 to demonstrate the difference in NDVI according to the season, as shown in Figure 4c. The tea NDVI showed differences not only temporally, but also spatially. Figure 4a,b shows the spatial variation of the tea NDVI-based vegetative cover in January and June. In general, in almost all years, the southern and northern parts of tea plantation were dense vegetation, denser than the center part. The spatial variation of NDVI between parts was due to the tea age. Meanwhile, the southern part includes Trung Dong, Than Thuoc, and the northern part is Phuc Khoa, which can be described as new tea plantations. On the other hand, the center part is less vegetation dense than the southern and northern parts. The center part is Tanuyen town, which has the biggest tea area. The result of the present study showed that the tea NDVI values

varied at each garden. This means that the NDVI value also has a difference at distinct monitoring points. Will the temperature have an impact on the tea NDVI value or not? We demonstrate this by establishing the relationship between temperature and tea NDVI in the next section.

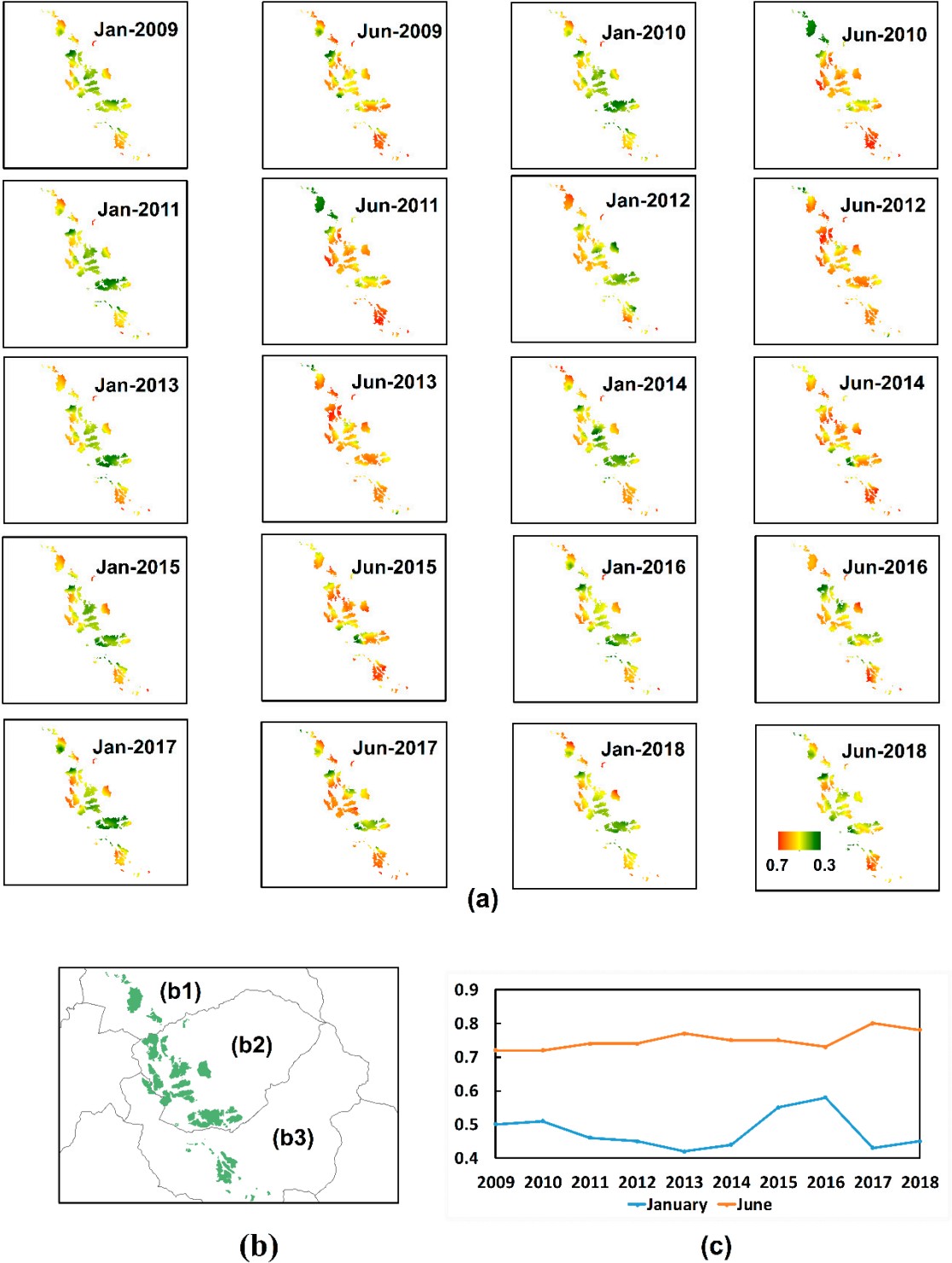

**Figure 4.** (**a**) Normalized Difference Vegetation Index value of tea in Tanuyen in January and June from 2009 to 2018 at each garden; (**b**) Tea garden location, (**b1**) Phuc Khoa, (**b2**) Tanuyen town, (**b3**) Trung Dong-Than Thuoc; (**c**) The monthly average.

The tea NDVI value increased from March to September, and obtained the highest value in July, August, and September. The tea NDVI value decreased from October to February of the next year, and obtained the lowest value in January and February. The mean NDVI value of tea changed each month from 2009 to 2018, as shown in Figure 5. The minimum and maximum average values of NDVI were calculated in the period from January 2009 to December 2018 (Table 3). The result of the study showed that the minimum NDVI value was 0.42 during January 2013 and February 2015 and 2016. The maximum NDVI values were in August 2015 and June 2017.

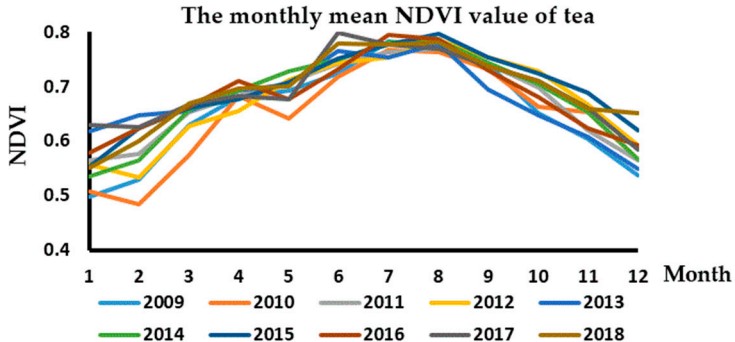

**Figure 5.** The graph shows the change in tea NDVI value each month from 2009 to 2018.

The USGS [41] suggests that "NDVI values range from +1.0 to −1.0. Areas of barren rock, sand, or snow usually show very low NDVI values (for example, 0.1 or less). Sparse vegetation such as shrubs and grasslands or senescing crops may result in moderate NDVI values (approximately 0.2 to 0.5). High NDVI values (approximately 0.6 to 0.9) correspond to dense vegetation such as that found in temperate and tropical forests or crops at their peak growth stage". Depending on the threshold, we proposed the class names as "moderate" and "healthy" for monitoring tea NDVI condition for the four seasons in a year (Figure 6), with a rank of 0.4 to 0.59 being moderate and ≥0.6 being healthy (high NDVI value). From 2009–2018, the number of months for "moderate" tea was 33 (27.5%) and "healthy" tea was 87 (72.5%).

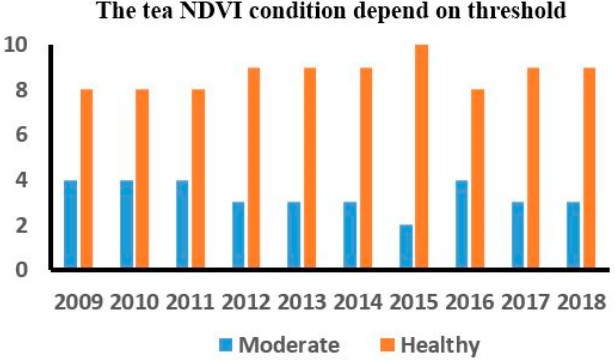

**Figure 6.** Statistics of the tea NDVI according to two thresholds "moderate" and "healthy" from 2009 to 2018. The red column is the number of months for "healthy "tea and the blue line is the number of months for "moderate" tea.

### 3.2. The Prediction of Tea Yield

#### 3.2.1. The Correlation between Climatic Variables, NDVI, and Tea Yield

The climatic variables and NDVI were considered as predictor variables in the TLRM, SVM, RF, model. Meanwhile, the climatic variables include mean temperature, minimum temperature (Tmin), max temperature (Tmax), precipitation, and solar radiation. To determine the impact of these variables,

we carried out regression models between the tea yield (predictand) and NDVI and meteorological factors (predictor variables), shown in Figure 7.

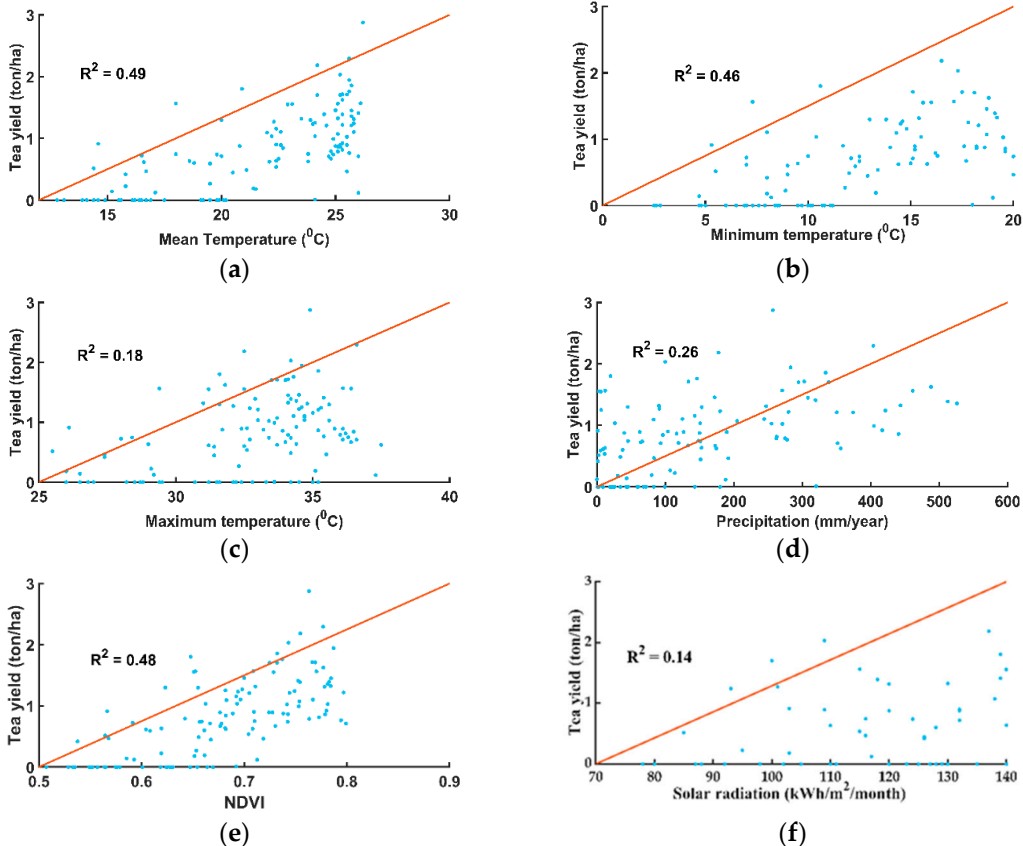

**Figure 7.** The relationship between tea yield, NDVI, and meteorological (linear relationship) period 2009 to 2018.(**a**) Description of the linear relationship between tea yield and the mean temperature; (**b**) Description of the linear relationship between tea yield and Tmin; (**c**) Description of the linear relationship between tea yield and Tmax; (**d**) Description of the linear relationship between tea yield and precipitation; (**e**) Description of the linear relationship between tea yield and NDVI; (**f**) Description of the linear relationship between tea yield and solar radiation.

Tanuyen District is located in the tropical monsoon climate, with cold winters in winter. Frost often occurs in December and January, which is the time that people cut down the tea, so there is no value of tea production in January and February. The effect of temperature on productivity was determined by regression analysis between the monthly mean temperature in the year and yield data (Figure 7a) including months without tea harvest (January and February). The trend line shows that the years that had the highest yield was at 26 °C. This temperature threshold is also equivalent to the temperature at which the tea reached the highest NDVI value. There were decreasing tea yields when there was a return to colder average monthly temperatures in November and December with the temperature around 17 °C. Observations from Figure 7b show that Tmin may affect tea yield more than Tmax (Figure 7c). The max temperature and yield may not be linear, because the $R^2$ was only 0.18. There may be a more complicated relationship between them, which needs further discussion. Past research has shown that the appropriate ecological temperature of tea is 18–23 °C, and at temperatures below −5 °C and over 35 °C, the tea will stop growing and be of less quality.

The precipitation effect on yield was determined by regression analysis between precipitation and yield data including the dry months. A few years had heavy rainfall in February, and the tea had buds to harvest in March in 2013, 2014, 2015, and 2017. The rainy season takes place from April to

September, which is also the time when tea bushes have their highest yield. Figure 7d shows that the optimum precipitation giving the highest yield was found to be around 300 mm in June and July.

Remote sensing applications have been used as an assessment and monitoring tool for the parameters of crop vigor, and yield estimation for crops during the past several decades [16]. The status and yield crop can be used for early forecasting and provide up-to-date information before the crop harvest period through the crop monitoring system. Tea NDVI was extracted from the NDVI of all provinces and tea boundaries. The mean NDVI was calculated by MATLAB software. Figure 7e shows the regression analysis of NDVI and the monthly mean tea yield. Note that the months with low NDVI values where the months of tea yield had a value of 0. This is usually the months of December, January, and February when the temperature is low, sometimes with frost, which have poor productivity so people cut down the tea (from 10–15 cm). In mid-spring, tea can start being harvested, at this time, the NDVI value was 0.55 and over, and the highest yield was always in July or August with a NDVI of 0.7 to 0.8. The correlation coefficient 'R' values between tea yield and its NDVI showed that the correlation is positive and significant [21].

Solar radiation is also one of the factors that affect tea yield as research has shown that solar radiation has a relationship with tea yield. The relationship is quite weak and significant, indicating that solar radiation all year round affects the tea yield, as shown in Figure 7f.

### 3.2.2. Predicting Tea Yield Base on Support Vector Machines (SVM) and Random Forest (RF)

The capability of SVM and RF models in replicating monthly mean tea yield was assessed by comparing the shapes of the actual yield and yield estimated by SVM and RF, as shown in Figure 8. To compare the accuracy of yield prediction, the line shapes of the actual yield and yield estimated by SVM and RF and the traditional linear regression model (TLRM) are represented. The blue line in the figures shows the monthly observed yield, the orange line represents the yield estimation of the SVM model (Figure 8c), and the red line represents the yield estimation of the RF model (Figure 8b). The yellow line represents the yield estimation of the TLRM (Figure 8a). The overall reaction of the model year by year had a good level of accuracy with the curves of the predicted, and the observed yields displayed the same pattern. This is especially true in 2015 by the three models. In general, the results show that the SVM and RF models were relatively similar to that actually observed, especially in 2012 and 2015 with a $R^2$ (coefficient of determination between actual yield and yield estimated by SVM and RF model) from 0.79 to 0.9 (Table 4). For the period 2009 to 2018, the predicted tea yield by the RF model was the best with a $R^2 = 0.73$. Meanwhile, high accuracy was seen in most years, with a $R^2 > 0.7$ (except for 2010, 2017, and 2018). The curves of the actual yield and yield according to RF had an approximate value. This can be explained because the $R^2$ by RF was the highest for the period 2009 to 2018. For the SVM model, the accuracy of prediction year by year was quite good, with a $R^2$ above 0.6 (except 2010 and 2018) and 0.66 for the total period. 2015 showed the highest accuracy, with $R^2 = 0.9$. For TLRM, the coefficient of determination between TLRM and actual yield was 0.57 for all years. Furthermore, the evaluation indicator $R^2$ was lower than 0.5 in 2009, 2010, 2013, and lower than 0.6 in 2011 and 2018; however, 2015 indicates a special case, with $R^2$ the highest by the three models. Nevertheless, using the coefficient of determination is not enough for the accuracy evaluation of the model, so two evaluation indicators (RMSE and PETY) were used in the next section.

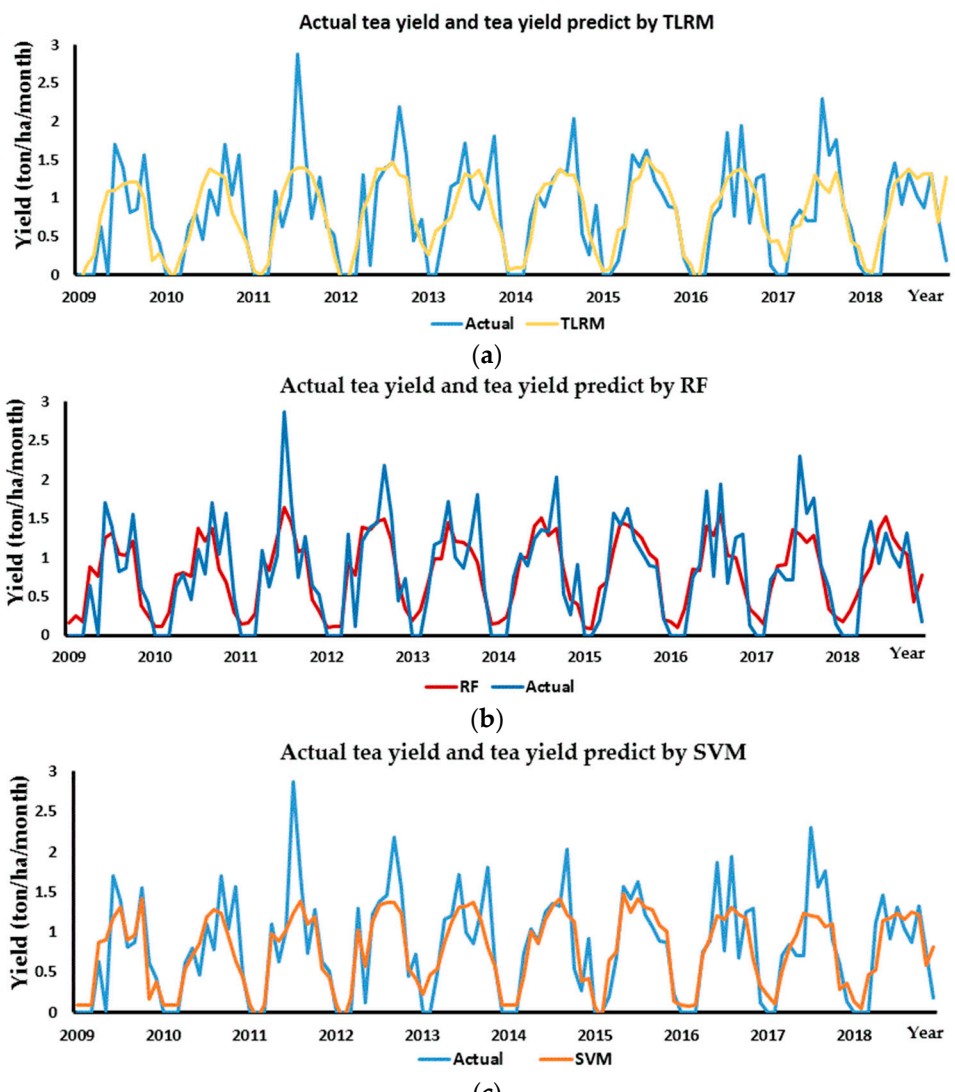

**Figure 8.** The performance of the three methods (Support Vector Machine, Random Forest, and Traditional Linear Regression Model) across different years to compare actual yield and estimation yield: (**a**) Actual yield and estimation yield by TLRM; (**b**) Actual yield and estimation yield by RF; (**c**) Actual yield and estimation yield by SVM.

**Table 4.** Coefficient of determination $(R^2)$ between the actual yield and estimation yield by SVM, RF, and TLRM.

| Year | $R^2$ | | |
|------|------------|-----------|-------------|
|      | **SVM-Actual** | **RF-Actual** | **TLRM-Actual** |
| 2009 | 0.69 | 0.71 | 0.4 |
| 2010 | 0.58 | 0.6 | 0.45 |
| 2011 | 0.66 | 0.8 | 0.59 |
| 2012 | 0.86 | 0.79 | 0.62 |
| 2013 | 0.57 | 0.74 | 0.49 |
| 2014 | 0.66 | 0.74 | 0.64 |
| 2015 | 0.9 | 0.87 | 0.89 |
| 2016 | 0.68 | 0.75 | 0.59 |
| 2017 | 0.61 | 0.67 | 0.61 |
| 2018 | 0.52 | 0.52 | 0.54 |

Scatter plots of the monthly observed and SVM, RF, and TLRM models for the period 2009–2018 at Tanuyen are shown in Figure 9. The figures show that the SVM and RF estimated yield was very close to the actual data collected by the Department of Natural Resources and Environment, Tanuyen District. On the other hand, the actual yield scatter plot and the estimated monthly yield of SVM, RF, and TLRM showed that RF was the best at estimating yield.

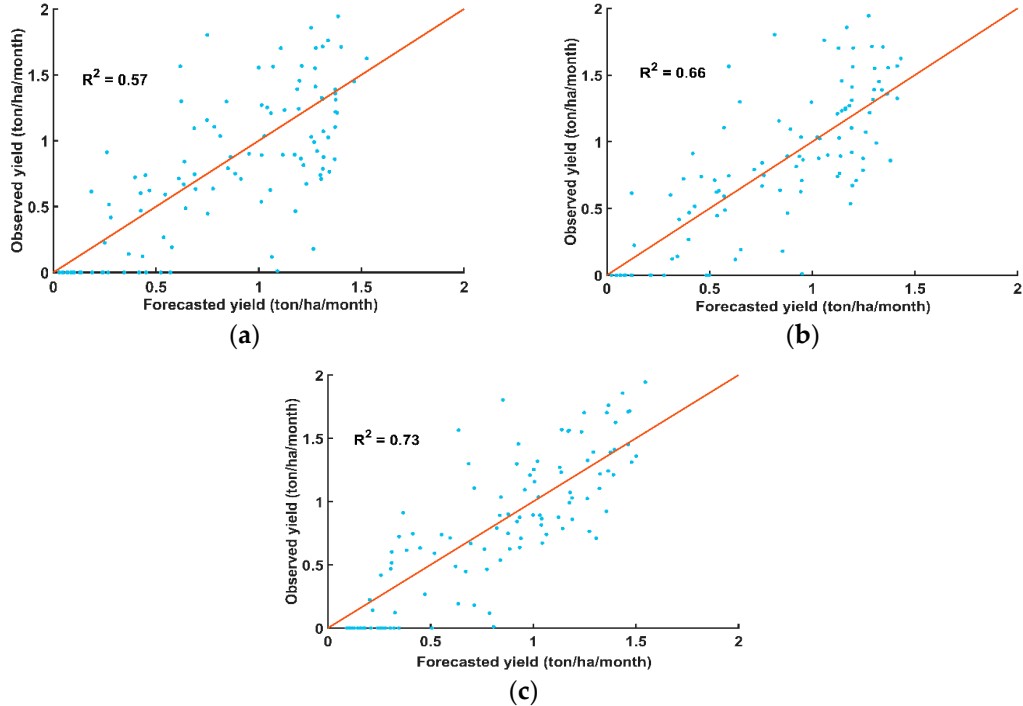

**Figure 9.** Scatter plots for monthly observed yield and forecast by SVM, RF, and TLRM from2009 to 2018 in Tanuyen District. (**a**) The observed tea yield and forecasted tea yield by TLRM; (**b**) The observed tea yield and forecasted tea yield by the SVM model; (**c**) The observed tea yield and forecasted tea yield by the RF model.

### 3.2.3. Correlations Analysis between Tea NDVI and Temperature

The climate was demonstrated to be closely related to the NDVI variability [48]. X. W. Chuai et al. showed that the temperature impact on NDVI was obviously more than precipitation [49]. Zhang et al. reported that the impact of temperature was significantly stronger than the impact of precipitation [50]. Therefore, in this study, we only used the effect of temperature on tea NDVI. Figure 10 shows that the increasing or decreasing trend of temperature can affect tea NDVI.

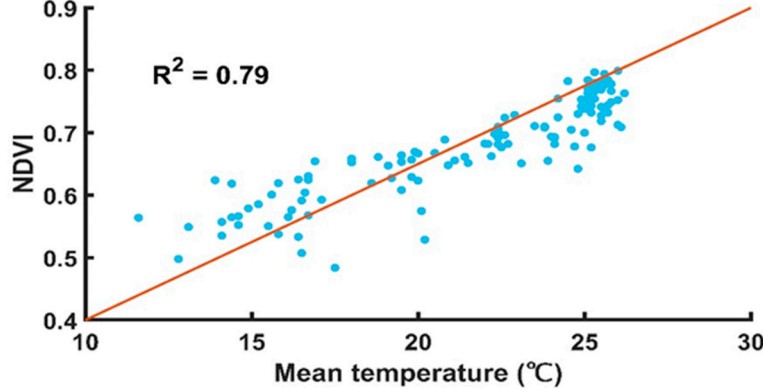

**Figure 10.** The relationship between the tea NDVI value and the monthly mean temperature 2009–2018.

The effect of temperature on the NDVI of tea from 2009 to 2018 was examined. According to the monthly mean NDVI combined with the temperature value of the same time, correlations between the monthly mean NDVI and temperature were analyzed, with a $R^2 = 0.79$. In this study, the linear relationship was chosen to indicate the correlation between NDVI (dependent-variable) and temperature (independent-variable). The NDVI values and temperature were plotted and linearly regressed. The relationship was strong and significant, which means that temperature strongly affects the NDVI value of tea. The average of the actual NDVI values and temperature were calculated and the linear regression analysis was carried out using NDVI values and temperature. The trend line on the effect of temperature and the value of the tea NDVI showed that optimum temperature for the highest NDVI value was found to be around 25 °C. These values will decrease when the temperature is below 17 °C. When the maximum of temperature in a month is too high (over 35 °C), the NDVI of tea will definitely decrease.

### 3.2.4. Comparison of Accuracy of Prediction Models and Forecast Errors

We plotted the RMSE and PETY by period from 2009 to 2018 to indicate the accuracy of the prediction of the three models (Figures 11 and 12). This section shows the results obtained after the implementation of the algorithms on the tea dataset with the SVM, RF, and TLRM models. The RMSE (Figure 11) suggests a discrepancy within the actual yield and forecast yield. The highest accuracy was indicated in 2015 with the smallest error by the three models (<50 kg/ha), regardless of the algorithms. Three models showed the highest accuracy in 2015 with less than 3% error. In the other years, the prediction accuracy was higher through the SVM and RF models. Meanwhile, the RF algorithm gave a better PETY (≤10%) and the root mean square error for this algorithm was significantly less (≤80 kg/ha). The RMSE and PETY was relatively good by the TLRM model with a RMSE from 80 to 100 kg/ha and PETY from 8 to 15%, except for 2015. The difference in the prediction result came from some uncertainties, mainly from data sources such as NDVI, resolution of the remote sensing, and meteorological data. The mean accuracy decreases, depending on the change in the variables. Correlation between tea yield and some of the variables are displayed in Figure 7.

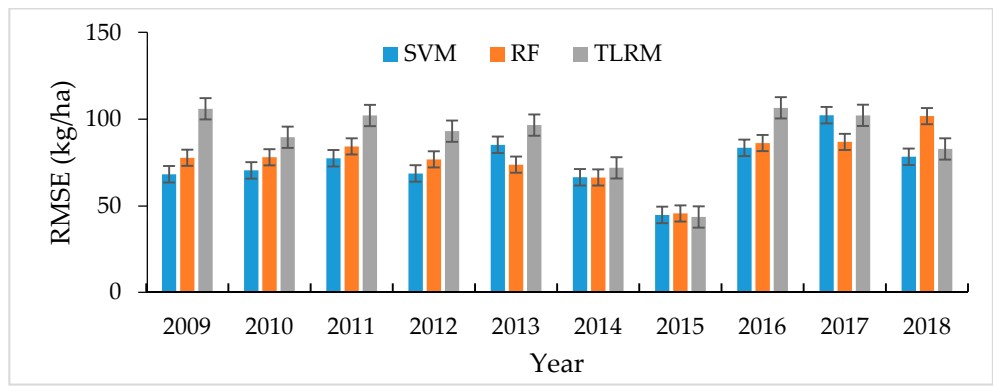

**Figure 11.** The RMSE (kg/ha) between tea yield forecasts and observations on the SVM, RF, and TLRM model period of 2009–2018. Error bars represent one standard error.

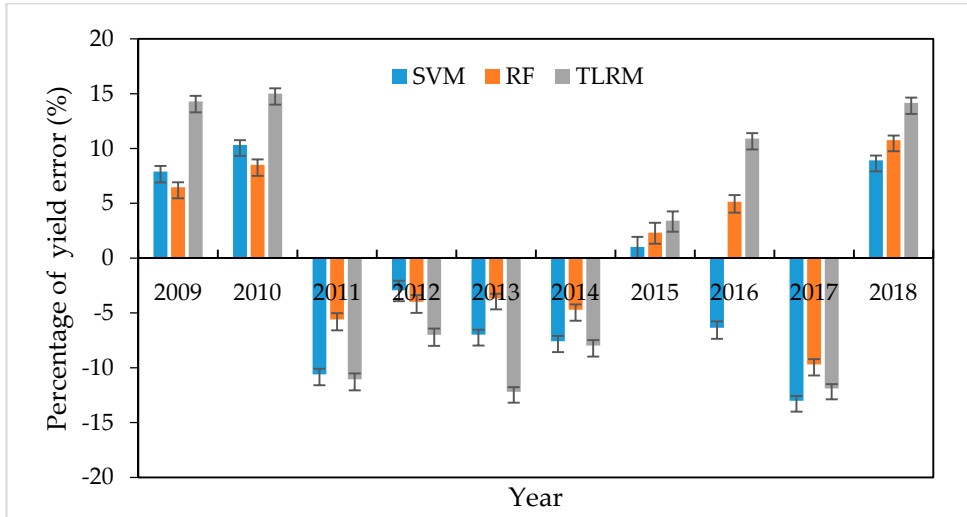

**Figure 12.** Percentage of yield error of tea yield in the period 2009 to 2018. Error bars represent one standard error.

## 4. Discussion

### 4.1. The Status Monitoring of Tea by NDVI

Numerous previous studies have demonstrated that NDVI is an effective tool in the assessment and monitoring of tea bushes, however, the temporal was limited. Meanwhile, tea health should be monitored for a long time. In our study, the tea status was monitored over 10 years, from 2009 to 2018 by extracting NDVI. Our analyses also showed that temperature is an important factor for tea growth and that there is a tea NDVI relationship to change in temperature. The average annual air temperature in Tanuyen is relatively cool, reaching an average of about 23 °C. There are two cold months (≤18) in January and December. May and June are the months with the highest air temperature, reaching 32.7 °C. This is consistent with our research. Our results showed that temperature was positively correlated with NDVI. The lowest NDVI value of tea was always in January (Table 3) when Tanuyen had the lowest temperature in the year. The highest NDVI value of tea was in June, July, and August, and this time period has the most suitable temperature for developing tea plants. The result indicates that (1) the change of NDVI in season was represented, as illustrated by January and June, which is also the coldest and hottest time in Vietnam. Figure 4a shows that the tea NDVI value of each garden had a threshold of 0.3 to 0.7. Then, the monthly average NDVI value of tea was calculated and illustrated in Figure 4c. The change of NDVI was demonstrated in the linear relationship with a temperature of $R^2 = 0.79$. This relationship was indicated in Figure 10. (2) The mean NDVI value of tea was found in each month through MATLAB software. Depending on the USGS [41], we proposed class names of "moderate" and "healthy" thresholds, where a rank between 0.4 and 0.59 was "moderate" and ≥0.6 was "healthy".

### 4.2. The Forecast Tea Yield

In the previous study, the prediction of tea yield was developed depending on the relationship between NDVI and leaf area index, or the relationship between NDVI and meteorological factor [24–28]. They were developed as multiple linear models to predict tea yield using climatic variables. There was a strong causal relationship between climate variables and tea yield [24,25,51]. Crown density of tea was computed based on NDVI and tea yield was computed based on crown density [23]. The linear relationship between NDVI and leaf area index (LAI) of tea was measured by Dutta [10]. Our results show that NDVI and climate data are crucial in predicting tea yield. The highest correlation values were observed with the mean temperature and NDVI, while solar radiation showed the least correlation

values. On the other hand, the correlation between NDVI and temperature was analyzed. Integrating the advantages of NDVI and climatic variables may contribute more to yield forecast. The correlation between NDVI–tea yield and climate–tea yield are relatively close. Furthermore, close correlations between NDVI and temperature were found. The combined NDVI and temperature may be better for predicting yields, which should be studied in more detail in the future. The results also showed that at least half of the observed events were correctly forecasted and thus the majority of the forecasts were true.

In our research, we attempted to use some regression models to predict the yield of tea based on changes in mean temperature, Tmin, Tmax, precipitation, solar radiation, and NDVI over the area of study. The traditional linear regression model, which has become obsolete, was carried out to predict the tea yield with $R^2 = 0.57$. We used machine learning methods as a way to improve tea yield prediction that has not previously been mentioned in research, but was successfully applied to predict crop yield.

The estimation showed that the yield predicted by the SVM and RF model was very similar to that actually observed. The curves of the actual yield and yield according to RF had an approximate value. In this circumstance, the RF model was more effective at estimating yield. On the other hand, the SVM method showed the highest accuracies of the linear relationship between actual yield and estimation yield by SVM with $R^2 = 0.9$ (2015).

### 4.3. Limitation and Future Perspectives

There are several limitations to this study. First, the uncertainty of MODIS NDVI data due to the effects of atmospheric, meteorological data, and satellite image quality were not considered in the model. Van Leeuwen et al. [52] confirmed that MODIS NDVI data were affected by atmospheric water vapor, even though this effect was minimal. In addition, the uncertainty, particularly in precipitation, and in general, meteorological have been addressed by Savino Curci et al. [53] and C.Merker et al. [54]. These uncertainties of data can develop in the near future when considering the error of various variable input in the prediction yield. Second, we used a Sentinel-2 image on 3 November 2018 to classify and establish the LULC map. As a general problem, the accuracy of classification may influence the subsequent analysis results, due to the survival of mixed pixels. After classification, the tea area was in 2018, meanwhile, the tea area changed in area from 2009 to 2018. Although this change was not significant, they may result in certain influences. Before 2014, we tried to establish the LULC each year by the LANDSAT 4–5 image, but they could not give a good result, so the Sentinel-2 in 2018 was chosen for the tea area. Therefore, to lift the accuracy of the method, various kinds of data sources should be used in more high-resolution images. Third, the NDVI and meteorological data were considered as independent factors to predict tea yield. Meanwhile, the factor affected by tea yield is significant including soil, $CO_2$, moisture deficiency, insects, etc., which can be developed in the future. Third, we explored the tea yield by the SVM and RF model, depending on the historic data over 10 years, however, the cause of the accuracy difference between each model was not investigated. Future studies may focus on the investigation of these problems with the help of higher satellite images and utilizing time-series machine learning techniques such as RNN (recurrent neural network).

Previous studies have also indicated the importance of the vegetation index and meteorological data in identifying and predicting tea yield. NDVI plays an important role in identification, assessment, and tea yield prediction [7–11]. The climatic variables were positively correlated with the tea yield [6,24–26]. Among the climate factors, temperature variability was determined as a stronger positive effect than precipitation in the predicted tea yield [43,51,54]. In our study, NDVI and mean temperature played a more important role than the other variables for estimating yield (Figure 8), which can be attributed to the different order as follows: mean temperature > NDVI > Tmin > precipitation > Tmax > solar radiation.

## 5. Conclusions

Tea is one of the main export products of Vietnam. Tanuyen District is one of the three key tea areas of Laichau Province. This is also a place with a long tea-growing history from the years of the 1960s. Tea trees also bring a major source of income for farmers in Tanuyen. Tea yield, however, has stagnated in recent years, due to changes in temperature, precipitation, the age of the tea bushes, and diseases. In this context, tea status monitoring and yield prediction are necessary. However, studies on tea are limited. Using multi-temporal remote sensing data, the paper details an investigation of the changes in tea status and yield estimation through the normalized difference vegetation index (NDVI) and meteorological data. The features of this study can be highlighted according to the following results. (1) The land-use/land-cover map was established to cover the tea area and the tea NDVI value was analyzed. The NDVI value of tea was extracted and we calculated the average to demonstrate the temporal and spatial changes in NDVI. The minimum and maximum average values of NDVI were calculated for the period from January 2009 to December 2018. The result of the study showed that the minimum NDVI value was 0.42 during January 2013 and February 2015 and 2016. The maximum NDVI value was in August 2015 and June 2017. On the other hand, the relationship between tea NDVI and mean temperature was demonstrated. (2) Support vector machine (SVM), random forest (RF), and the traditional linear regression model (TLRM) were the three models used to predict tea yield and we examined the relationship between tea yield and six factors: mean temperature, Tmin, Tmax, precipitation, solar radiation, and NDVI. For the period 2009 to 2018, the prediction tea yield by the RF model was the best with a $R^2 = 0.73$, with 0.66 in SVM and 0.57 in TLRM. The highest accuracy for the three models was 2015, with $R^2 \geq 0.87$, RMSE<50 kg/ha, and PETY with less than 3% error. In the other years, the prediction accuracy was higher in the SVM and RF models. The RF algorithm gave a better PETY (≤10%) and the root mean square error for this algorithm was very significant (≤80 kg/ha).

Using multi-temporal MODIS NDVI data is inexpensive and is a useful method for monitoring tea status and yield estimation. Through machine learning approaches such as SVM and RF, these methods demonstrated that they are extremely effective in the improvement of yield prediction and can be used widely to estimate tea yield in the future.

**Author Contributions:** Conceptualization, P.P. and L.X.; Methodology, P.P., L.X., and N.C.; Software, P.P. and L.X.; Validation, P.P, L.X., and Z.C.; Formal analysis, P.P.; Investigation, P.P.; Resources, P.P.; Data curation, P.P; Writing—original draft preparation, P.P.; Writing—review and editing, N.C., Z.C., and L.X.; Visualization, P.P.; Supervision, N.C.; Project administration, N.C. All authors have read and agreed to the published version of the manuscript.

**Funding:** Funding for this project was provided by the National Key R&D Program (2018YFB2100500), the National Natural Science Foundation of China Program (41890822), Creative Research Group of Natural Science Foundation of Hubei Province of China (2016CFA003), the Fundamental Research Funds of the Central Universities (2042017GF0057), and the National Nature Science Foundation of China Program (41771422, 41971351, 41601406, and 41801339).

**Acknowledgments:** The authors would like to thank the hydro-meteorological station of Laichau Province and Department of Natural Resources and Environment, Tanuyen District, Laichau Province, Vietnam, which provided the data for this study.

**Conflicts of Interest:** The authors declare no conflict of interest.

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
