# Peer review of "Using Multi-Temporal MODIS NDVI Data to Monitor Tea Status and Forecast Yield: A Case Study at Tanuyen, Laichau, Vietnam"

_remotesensing, doi:10.3390/rs12111814_

Round 1
Reviewer 1 Report
I can see a lot of potential in this paper but serious editing should be considered by the authors before resubmitting this manuscript. Also, the authors have some serious methodological issues in the understanding and use of Machine Learning algorithms and linear regression models. I do not want to discourage the authors by my comments but this paper is not ready for submission yet. Please, start all over again and pay more attention to the methodology.
Particular comments:
Line 17: the paper details…
Line 52: ‘…and NDVI which ARE the most important factors AFFECTING TEA YIELD.’
Line 51 to 53: please clarify verb tenses: future, past…
Line 52: vegetation condition (singular)
Line 56: TO provide…
Line 60: ‘It mean, stressed plant was displayED dark amongst…’
Line 63-4: please clarify ‘temporal’.
Line 67: tea class classification [10-12]
Line 69: ‘was assessed..’ DELETE: ‘…and are effective tools.’
Line 71: ‘texture’ is a vague term. Please, clarify what you mean by texture.
Line 97: two dots before Dutta[9]
Line 97: check the verbs tenses.
Line 101-102: please tidy up this sentence.
Line 104: what is wrong with the TLRM?
Line 106-111: so what? What happened to these Machine Learning approaches? Were they good? What was the performance in comparison to TLRM?
Line 113-115: ‘To improve the prediction accuracy of tea yield, this 113 work proposes the prediction model for yield forecast based on Support Vector Machine (SVM) and 114 Random Forest (RF) to replace the traditional linear regression model (TLRM).’ Why? Can you unpack your reasons?
Line 122: separate 10 m.
Line 123: future tense cannot be used in a paragraph full of past tenses…
Line 126-9: revise the sentence
Line 141: were selected.
Line 142: imageS HAVE
Line 144-6: Amend the sentence…
Line 147: Define small fraction of cloud cover
Line 157-159: I do not understand what you mean in this sentence. The following sentence does not seem to fit in the line of the argument.
Line 162: ‘The other data THAT were used for…INCLUDED…’
Line 179: ‘The climate is rainy and dry season is quite clear.’ This seems to contradict each other.
Line 189-90: you mean registered, georeferenced to a common proyection?
Line: 191: APPROACHED
Line: 192-3: that is fine but why??? What were the considerations for taking that particular decision? Was it absolutely necessary? You have to provide a paragraph somewhere to justify the use of these methods and not others. Please, explain were the decision came from.
Line 195-6: you already said that.
Line 196: IN Erdas Imagine…
Line 198: SEPARATE
Line 198-9 : you already said in line 165-6 that you were using a LULC classification. So, why are you classifying this again?
Line 206-7: the sample points were arbitrarily chosen by you using a manual photointerpretation or did you use a particular system? Why not using the ground truth data or part of it?
Line 210-11: I do not follow what you mean by raw and smooth texture…
Line 215-6: I do not understand this sentence.
Line 222: Who was geometrically corrected?
Line 223: you resampled a MODIS image to 10m??? Really? How?
Line 221-226: Please, tidy up all these sentences. Also, how are the NDVI values being affected by the process of resampling and then averaging them within the boundaries of the polygons that represent the tea plantations?
Line 228-9: why did you do a linear regression between NDVI and temperature? What for?
Line 232: were USED: Support…
Line 233-4: The TLRM was used to compare the accuracy of the TLRM???
Line 236: WERE compared…
Line 240-2: who said that?
Line 266: ‘THE accuracy of SVM and RF models ARE DEPENDANT depending mainly on THE selectED the model parameters.’
Line 266-268: I can see a few autocorrelations in these data sets: e.g. Temperature and NDVI, NDVI and historic tea yield… Besides, you are trying to predict tea yield with historical tea yield???
Line 270-1: I do not understand this sentence.
Line 321: Please tidy up this sentence. I know what you mean but it does not sound right.
Line 321: The NDVI HAS…to WITH…
Line 323-4: WAS SUCCESSFULLY USED IN… ASSESSMENTS…TO DETECT
Line 338: simply say IN WINTER.
Line 346: MEANWHILE
Line347: DESCRIBED…
Line 349: ARE DIFFERENT AT…
Line 351: BY ESTABLISHING..
Line 352-3: please tidy up this sentence. It does not make sense. Do you refer to the phenology of the tea plants?
Line 355-6: It is not very clear to me why the discrepancies are necessarily related to tea age. You seemed to imply that variations were related to time of the year… Can you clarify this point?
Line 368-372: I do not follow this classification… and the consequences of it.
Line 378-80: did you also include NDVI as a climatic variable?
Line 385-7: if you did not have tea harvest then why using these values to predict the effect of temperature with productivity?
Line 388: not so clear this statement… below the maximum yield you can find a lot of values that are similar with lower mean temperatures.
Line 391: Possibly because at higher temperatures plants close stomata activity to avoid structural damage whereas in the winter this is not a theat. So, the relationship between TMax and yield is not linear.
Line 396-7: so, if rain is seasonal why using average rainfall and not total within the rainy months?
Line 397-8: Figure shows precipitation per year… NOT in June-July… Please, clarify this.
Line 407-8: which NDVI values did you use, spring, summer???
Line 412: solar radiation when? All year round, a particular month…???
Line 427: what were you using as an independent variable in the TLRM to estimate yield? Then, explain how TLRM and RF/SVM are comparable? Are they using the same data?
Line 472: you should use accumulated temperature and NDVI more than individual values. If NDVI is related to the vigour of the vegetation and this is a product of accumulated growth during a season. Mean temperature may be a good estimate but not necessarily the best one. Also, temperature dissociated to water availability may not be giving you accurate results. This LR is not relevant as it is.
Figure 11: y-axis lacks scale.
Line 515: ‘Numerous previous researches were demonstrated…’
Line 518: ‘…an important affect factor…’
Line 540: where did you measure the relationship between NDVI and LAI????
Line 544-546: by combining NDVI and temperature you necessarily had a high level of autocorrelation in your RF and SVM in your analysis. This is not right.
Line 550-1: this is not true. You used a multivariate analysis in RF and SVM whereas the TLRM was bivariate. It is not comparable and even less conclusive.
Reviewer 2 Report
I think that the manuscript was adequately revised. Redundancy was improved, and the tables and graphs became informative dramatically.
Although this study became more suitable for publication, I expect that the authors will improve minor points mentioned below.
L65-66. The sentence like "... and the difference from previous research, the monitoring health of tea in Tanuyen should be made." may be appropriate because this is the Introduction part and your study will be introduced hereafter.
Table 4. Please clearly show that the values are R2. Description on Tables and Graphs should be informative without manuscript.
L488-489. Is this regression analysis different from an regression analysis conducted in Fig. 10? It is unclear for me. If so, please add explanation. Anyway, the mean temperature in this study appears less than 30 (Fig. 7 and 10) and an inference on over 35 degrees Celsius situation from the data should be difficult.
Fig. 12. Please show numerical values on the vertical axis of RMSE.
L519. English of this sentence is difficult to understand. Does it mean " Our analyses also show that tea NDVI have a relationship to temperature."?
L593. If considering both NDVI and meteorological variables is the strong point of your study, please mention about it after "studies on tea are limited."
Reviewer 3 Report
Thank you for taking the time to address the comments. Good Job.
Author Response
thank you so much!
Have a great day!
Round 2
Reviewer 1 Report
The authors have produced valid answers to the questions raised in previous reviews. I think the paper is a piece of honest work and I appreciate the hard work and the patience of the authors with the always lengthy process of reviewing. I am happy with the publication, although I am not very convinced this paper is making a clear case for the use of machine learning algorithms compared to the use of normal regression models. Machine learning can be a powerful tool to process large datasets and to discover new knowledge. However, scientists should always endeavour to explain what those tools are finding, which is something not evident in this manuscript. The authors are happy with the thesis that ML will produce better results than LR, which may be possible, but they do not explain why. The authors have not explained what kind of new knowledge ML algorithms found that the regression models could not.
On the other hand, I recommend the authors to review the use of the English language in this manuscript. I did my best to correct some sentences and to ask for clarifications but still it would be better that the authors try to amend the text with the help of a native English speaker.
This manuscript is a resubmission of an earlier submission. The following is a list of the peer review reports and author responses from that submission.
Round 1
Reviewer 1 Report
I have read the paper several times and I think the work has good grounds for a future publication. The subject is very interesting and potentially can be converted into a good tool to assist tea growers in Vietnam. However, I believe this paper needs further work before being resubmitted. As it is now, the work being described falls short of scientific interest: just another instance of NDVI usage. I recommend the authors to review the work of Reed et al., 1994 or Nellis et al., 2009 (https://www.researchgate.net/publication/228768339_Remote_Sensing_of_Cropland_Agriculture).
The use of the English language is rather poor in many sections with many spelling mistakes, wrong use of words and verb tenses, vague statements and confusing statements. Additionally, the paper needs to be structured in sections. Pieces of information that should be in the introduction appear in the discussion, new information appears in the conclusions, there are some repetitions in the Materials and Methods and the Results. An internal review of this paper could have improved the manuscript considerably before submission to this journal.
The methodology at simplicity, which is good. However, it is not very clear what the data represents, whether the interpretation of the results is correct and whether the methods have been correctly applied. It is very difficult to understand where the data is coming from and what the models are trying to represent. Perhaps a good idea for further analysis is to represent the variability of estimates within each production unit and comparing the results against the field measurements.
SPECIFIC COMMENTS:
lines 35-38: Amend the description
lines 47-8: add reference. On the other hand, your conclusions seemed to be different from this statement.
line 49: missing verb
line 55: plant(s).
line 56: display(ed)
line 69: change(d) and also add a dot before The tea...
line 76: 83%+27% equals 110%???
line 77-82: please review all this paragraph.
line 104-5: two or four machine learning approaches?
line 111: regress(ion).
line 126: specify what kind of data you were collecting and how.
table 1. How did you use base map data and field data? What are these datasets?
line 134-137: so what? why do you mention these problems with Sentinel 2A
line 139: you do not use time-series of Sentinel-2 images in your paper. Why do you say in here that you do? You only use an image captured on the 13th of November 2018. This is not a time-series.
line 149-152: I do not understand what you mean.
line 158: tea yield measured in what?
line 169: can you use latin names for these varieties of tea?
line 179: (such) as...
line 180: why are you doing this? can you provide at least a sentence that explains why you decided to do this comparison?
Section 2.2.1: did you do any atmospheric correction in the preprocessing? what kind of ESA product did you used? Also, you start Section 3.1 repeating information already stated here. Please, tidy up these two parts.
Section 2.2.2: can you please explain the aggregation process of NDVI. I guess, you used MODIS (not clear in the text), and then you aggregated the pixels within each polygon representing tea plantations in order to give monthly values for the entire area? Could you clarify that? This section also includes some statements that will be repeated again in the Results section.
line 237: it should say introduced by Breiman...
line 237-245: add reference
line 274: please review 'habit'
line 288: inconsistency. Sentinel-2 was not used to calculate NDVI
line 307: review sentence and add reference.
line 309-311: so what?
line 320-321: which values did you use for this classification, winter or summer?
section 3.2.1 What you mean by factor? you mean variables? As NDVI and temperature show a strong correlation, what is the point of using both of them? How did you calculate temperature? do you mean average temperature for the whole area or an aggregation of temperatures estimated at each location? You mention solar radiation but this variable does not seem to appear in Figure 8. Also, how did you measure this? You comment results in line 281 but I can see no graph, no results, nothing.
section 3.2.2 you seem to use temperature,solar radiation and NDVI to predict yield. First of all, this is not a good idea because temperature is strongly autocorrelated with NDVI. You are overfitting the model. Secondly, you do not describe where the yield values are coming from. Third it is not very clear whether you are predicting regional values, local values at each plantation, at each pixel, etc. As you are using MODIS pixels that are 250m size, you should at least comment how you are dealing with boundary or mixed pixels and how they are likely to alter results.
Figure 10. I do not know what is this representing. Total period, a year in particular, a location, the entire district???
Table 4. you focused on the good results for SVM in year 2015. However, you also have similar results in the LR model for that year.
4. Discussion: line 442-456 all this text should not be in the discussion section. It belongs to the introduction. Same for 468 to 472
Line 483-484: if this was not mentioned before why did you refer to the literature [27-32]. It seems contradictory.
Line 489. The difference between .90 and .89 does not seem highly relevant. However, any idea why this year was so good in all these methods compared to previous and all the years after this date? It makes me think that the method in not robust enough as it seems highly dependant on the quality of the images. Any comments on that?
line 495-496: can you explain why Landsat was not good and Sentinel better? Later you explain that high resolution images may do the trick better BUT, at the end of the day you base your analysis on MODIS, which is lower resolution than Landsat or Sentinel. This does not make any sense at all.
Line 504: RNN, why you did not use it if you seem to believe it may perform better?
Conclusions: line 524 why 2009 and 2015 precisely? by looking at Table 4, I can see that 2012 to 2016 was far better for RF. This method by the way seems more consistent than the SVM that appears to have lower values than RF 7 out of the 10 years being analysed. Also the interpretation of the results is not very rigorous. Can you tell anything about the residuals, any bias, scedasticity, outliers... ?
Reviewer 2 Report
This study reports the result of prediction of tea plantation and tea yield using satellite data in a part of Vietnam. This methods and result were interesting, beneficial, and kindly described. However, several points should be improved to make the paper more suitable for publication.
First, the novelty and transferability of the study appears to be a little bit ambiguous. Application of SVM and RF to tea yield prediction may be a novel point. If so, it should be clearly remarked. In addition, I think the comprehensiveness can be originality of the study. That is, predicting not only tea yield, but also distribution of tea plantation using freely available satellite data (Sentinel-2 and MODIS) can be a strong point of this study.
Second, this script includes many duplicating and redundant parts. It should be rearranged and be more concise. Although some redundant parts were shown in the specific comments below, authors should check and correct the manuscript thoroughly from the view point of busy readers.
L64-76, L113-117. These sentences are a part of summary of the result and should not be presented in the Introduction part. Please remove.
Table 1. Please show the location (Lat and Lon) of the Hydro-meteorological station and clearly mention that the data was obtained from one point.
L196. In the study region, can tea plantation be easily from bushes and other croplands from the Google Earth? Please describe key visual characteristics of the tea plantation on the Google Earth and showed that no similar land cover type exists in the study area.
L209. Please show that the values of monthly mean NDVI of the whole study area were used for the consequent analysis. Spatial variation of the NDVI was averaged and not considered after Figure 4. The prediction was conducted on temporal variation of NDVI. This should be clearly described.
Fig. 2 Although this figure is informative, some points are confusing and should be improved.
First, some processes which are not parallel are shown as they were simultaneously conducted.
For example, “Pre-processing ”,”Classification”,”Accuracy assessment”,“Final land-use/land cover” should be connected in series. Second, “Mean tea NDVI values” should be connected the processes below by any means. Nesting boxes may improve the description.
L257-263, 284-289, 315-321. This part is redundant and should be removed.
L326-346. What the purpose of the analysis? The result was not discussed in the discussion part. It may be possible that discussing that the correlation between NDVI and temperature can bring the better performances of SVM and RF, which can consider complex relationships between variables. If so, this analysis should be moved to after the 3.2.2. If this result are not important for the main story of the paper, this and related parts should be removed.
L369-373. This part is redundant and should be removed.
L 375. What is this “O” ? Do you mean “0” ?
L380-382. Why do not you show the graph of solar radiation in Fig. 8?
L390-410. Please move this part to the Method part. In addition, the size of data set should be described. (the size of the training data set was 80% of 120 values = 96?)
L490-504. Please mention about generality or transferability of the study. The title says this study is “a case study”. Readers may have an interest in other cases.
Reviewer 3 Report
In this research, three popular models were used to assess tea status and forecast yield - Support Vector Machine (SVM), Random Forest (RF) and the traditional linear regression model (TLRM). Although the idea is very good and this kind of approach is important for predicting tea yield, the manuscript could be improved before accepted for publication. Below are my suggestions/views for improvement.
Line: 83-92:
Here rice, wheat, and potato were discussed, also consider discussing biomass/bioenergy crops. For instance, the NDVI concept in land use and land cover data layers were already used to simulate yield for bioenergy crops (Foster et al., 2012; Singh and Saraswat, 2018).
Line 104:
Please mention all 4 models.
Line 103-117:
The 3 models you selected were based on the fact that these models were used for corn. Are you trying to say corn is similar to tea? Has any of these models ever used for predicting tea yield?
Using R2 alone as an indicator for the performance of the model might be questionable. Have you used any other stat to compare the performance of the models? Please look at Moriasi et al, (2007) and similar papers for additional reference.
The performance of the model can be better judged by the uncertainty in the model due to varying inputs. Have you done any uncertainty analysis? Please look at Van Leeuwen et al., 2006; and similar papers for additional reference. If for some reasons, the uncertainty work is outside the scope of the study [though it should not happen], add a paragraph citing relevant information and mention what can be done in the near future.
References:
Foster, A. J., Kakani, V. G., Ge, J., & Mosali, J. (2012). Predicting biomass yield in bioenergy crop production systems using canopy NDVI. In Proceedings from Sun Grant National Conference: Science for Biomass Feedstock Production and Utilization, New Orleans, LA. Retrieved from www. sungrant. tennessee. edu/NatConference.
Singh, G., & Saraswat, D. (2016). Development and evaluation of targeted marginal land mapping approach in SWAT model for simulating water quality impacts of selected second generation biofeedstock. Environmental modelling & software, 81, 26-39.
Moriasi, D. N., Arnold, J. G., Van Liew, M. W., Bingner, R. L., Harmel, R. D., & Veith, T. L. (2007). Model evaluation guidelines for systematic quantification of accuracy in watershed simulations. Transactions of the ASABE, 50(3), 885-900.
Van Leeuwen, W. J., Orr, B. J., Marsh, S. E., & Herrmann, S. M. (2006). Multi-sensor NDVI data continuity: Uncertainties and implications for vegetation monitoring applications. Remote sensing of environment, 100(1), 67-81.